# Heart and Left Anterior Descending Coronary Artery (LAD) Exposure from Hypo-Fractionated Whole Breast Radiotherapy with a Prone Setup

**DOI:** 10.3390/cancers17091562

**Published:** 2025-05-03

**Authors:** Fabiana Gregucci, Elisabetta Bonzano, John Ng, Fereshteh Talebi, Maahi Patel, Dakota Trick, Sharanya Chandrasekhar, Xi Kathy Zhou, Maria Fenton-Kerimian, Ryan Pennell, Silvia C. Formenti

**Affiliations:** 1Department of Radiation Oncology, Weill Cornell Medicine, New York, NY 10065, USA; fgr4002@med.cornell.edu (F.G.); jon9024@med.cornell.edu (J.N.); fet4007@med.cornell.edu (F.T.); map4044@med.cornell.edu (M.P.); dat4015@med.cornell.edu (D.T.); shc2043@med.cornell.edu (S.C.); mak9186@med.cornell.edu (M.F.-K.); rtp9003@nyp.org (R.P.); 2Department of Radiation Oncology, IRCCS San Matteo Polyclinic Foundation, 27100 Pavia, Italy; e.bonzano@smatteo.pv.it; 3Division of Biostatistics, Department of Population Health Sciences, Weill Cornell Medicine, New York, NY 10065, USA; kaz2004@med.cornell.edu

**Keywords:** prone, LAD, heart dosimetry, cardiotoxicity

## Abstract

This study investigates how the position of a breast cancer patient during radiotherapy affects the amount of radiation received by the heart and the left anterior descending coronary artery (LAD), which are at risk of damage during breast cancer radiotherapy. The authors report their technique for breast radiotherapy with a face-down position (prone) and compare the radiation exposure to the heart and LAD with those achieved in patients treated with other techniques. The results show that the prone position significantly reduces the radiation dose to these critical organs, making it a safer approach. The proposed approach is easily applicable worldwide. By reducing the risk of heart damage in patients with left-sided breast cancer, it could ultimately benefit their long-term health while still effectively treating their cancer.

## 1. Introduction

Radiation therapy (RT) is an essential component of breast cancer treatment. Considering the high prevalence of long-term survivors, the risk of late treatment-related cardiac toxicities is a relevant concern, particularly in patients with multiple risk factors [1,2]. Darby et al. established a linear correlation between the mean heart dose (MHD) and a 7.4% risk of major coronary events for every additional gray (Gy) of exposure, with no minimum threshold [3]. The proximity of the left anterior descending coronary artery (LAD) to the chest wall and the need for radiation fields to include the entire mammary tissue in left-sided breast cancer carriers render the LAD a key structure at risk of exposure and potential late treatment effects [4]. Van den Bogaard et al. suggested that the MHD may not sufficiently predict cardiotoxicity, emphasizing the need for specific dose metrics for cardiac substructures, including the dose to the LAD, mostly in the presence of atherosclerotic plaques [5,6]. Tagami et al. reported a significant correlation between LAD dose and the risk of developing coronary stenosis ≥ grade 3 in women receiving left-side breast cancer RT, with a threshold for the LAD mean dose (Dmean) of 2.9 Gy [7]. Similarly, in a series of 375 irradiated left-side breast cancer patients with a median follow-up of four years, Zureick et al. identified an LAD Dmean of 2.8 Gy and an LAD maximum dose (Dmax) of 6.7 Gy, as the thresholds for significantly increased risk of any cardiac events [8].

However, despite this growing body of literature on the importance of LAD dosimetry, no consistent dose–volume–effect correlations have been reported. Detailed dosimetry studies based on systematic contouring of the LAD in large cohorts of patients are lacking. Because of the key role of coronary vessels in heart failure, minimizing heart and LAD doses are crucial mandates for modern adjuvant RT of breast cancer to reduce the risk of cardiac mortality. Several techniques, such as prone irradiation, deep inspiration breath hold (DIBH), and proton therapy in the supine position have emerged as promising techniques to achieve this goal.

Over the years, our research team has developed a specific cardiac-sparing protocol for breast irradiation based on a specific prone treatment setup [9,10,11,12,13]. We describe the setup in detail in the Materials and Methods Section (see Section 2.2). The key steps were as follows: (1) ensuring that the sternum aligns with the internal edge of the immobilization device to maintain a horizontal chest position; thus, preventing sinking or axial rotation (Figure 1), (2) placing the medial edge of the tangents at least 2.5 mm from the contoured LAD [12], (3) accurately contouring the heart surface and LAD. The prone setup leverages the effect of gravity to create a physical gap between the RT target and the contents of the chest (the lungs and the heart) while drastically reducing the chest wall excursion movement of breathing. The latter assures intra-fraction immobilization (during daily delivery of the radiation dose) as demonstrated by the video recording of dose delivery to the breast by the MR-Linac, in the prone position (Appendix A). Importantly, prone breast irradiation is easily reproducible and can be implemented without requiring the acquisition of expensive RT systems.

In this study, we describe the heart and LAD dosimetry, focusing on the MHD, LAD Dmean, and LAD Dmax of a consecutive series of 524 left-side breast cancer patients treated at a single institution with a regimen of prone hypo-fractionated whole breast radiotherapy (WBRT) with a concomitant boost to the post-operative cavity [10].

## 2. Materials and Methods

### 2.1. Patient Selection

Weill Cornell Institutional Review Board approved the retrospective analysis of dosimetry and clinical data for a consecutive series of all patients with left-side breast cancer who underwent breast conservative surgery, followed by adjuvant hypo-fractionated WBRT with a concomitant boost to the post-operative cavity from 2016 to 2023 (IRB number 23-12026826). Patients with right breast cancer and patients treated with different fractionation regimens or different radiation fields were excluded from this analysis. The final sample size consisted of 524 cases.

### 2.2. Treatment Simulation and Verification Procedures

Patients were simulated and treated prone with the index breast positioned within the opening of the dedicated prone breast positioning device (kVue Access 360 G2 board, QFix System, LLC, Avondale, Pennsylvania, USA), as described before [9,10,11,12,13,14]. In brief: Special attention is given to ensure that the sternum is placed on the internal edge of the prone breast board (covered by a thin layer of memory foam), to keep the chest horizontal and prevent “sinking” and axial rotation. The arms are positioned above the head, holding the handlebar. A customized vacuum cushion supports the arms and is used at each treatment to reliably reproduce the position during treatment. Before the scan, three radiopaque fiducial markers are placed on the patient’s back and sides, aligned by the laser lights from the opposite walls and the ceiling of the simulation room, to define a reference plane through triangulation, preventing translational and rotational variance. Free-breathing computed tomography (CT) (2.5 mm thickness) chest images are acquired. Once the images are approved by the treating radiation oncologist, the markers are replaced by permanent tattoos, utilized as the daily references for position and treating the patient prone, in the treatment room. In 2020, a system of surface-guided RT (SGRT) during simulation and treatment procedures was added to the tattoos. Cone beam CT and portal images are obtained weekly to verify accuracy of positioning.

### 2.3. Definition of Planning Volumes

The treatment volumes were defined based on the CT simulation imaging. For the definition of the whole breast target volume, the clinical target volume (CTV) was considered coincident with the breast tissue, consistent with the RTOG Breast Cancer Atlas for Radiation Therapy Planning 2009. To define the planning target volume (PTV), named PTV-breast, a 1.5 cm uniform margin was applied around the CTV to account for uncertainty and setup error [13], with editing, where necessary, to maintain a minimal distance of 5 mm between the treatment volume and the external surface of the rib cage and the skin surface. The post-operative cavity was identified using surgical clips and the visible post-lumpectomy seroma region. When necessary, reference to mammography or breast MR informed the definition of boost volume. The contoured post-operative cavity was expanded uniformly by a 1 cm margin, ensuring a minimal distance of 5 mm inside the contoured breast. This expanded volume, named PTV-boost, was subject to the same physical constraints as PTV-breast. The lungs, spinal cord, contralateral breast, heart surface, and LAD were contoured.

### 2.4. Treatment Planning, Dose, and Fields Definition

The prescription dose to the whole breast was 40.5 Gy, with the boost region (PTV-boost) prescribed to receive 48 Gy in 15 fractions (2.7 Gy/fraction plus a concomitant boost to the post-operative cavity of 0.5 Gy/fraction) delivered from Monday to Friday for 3 weeks [9,10]. The plan was to deliver 40.5 Gy to cover more than 95% of the PTV-breast and 48 Gy to cover more than 98% of the PTV-boost [9]. Three-dimensional tangent fields, including the PTV-breast, and the target coverage was achieved by placing the posterior edge of the field on a plane connecting the midline to the anterior extent of the latissimus [13]. In a previous report, we demonstrated how, by placing the medial edge of the tangent fields at least 2.5 mm from the closest point of the contoured LAD, LAD Dmax could be kept < 10 Gy and LAD Dmean < 3.3 Gy [12]. Consistently, we kept the medial edge of the tangent fields at least 2.5 mm from the closest point of the LAD. The LAD Dmax and the LAD Dmean doses here reported are consequences of this constraint.

### 2.5. Data Collection and Statistics Evaluation

The primary endpoints of this project were (I) to measure the MHD (Gy), LAD Dmean (Gy), and LAD Dmax (Gy), (II) to compare our MHD, LAD Dmean, and LAD Dmax results to those reported in the literature for RT with different techniques. The secondary endpoints were to collect the following cardiac and target dosimetry and volume data, including the LAD and heart doses received by 2% of volume (Gy) (LAD D2% and heart D2%) and heart Dmax (Gy): PTV-breast V95% (%), PTV-boost V95 Gy (%), PTV-boost Dmax (%), breast volume (cc), and post-operative cavity volume (cc). The treatment planning system (TPS) used to perform dose calculation was Eclipse TM (Varian Medical System, Palo Alto, CA, USA). For each patient’s plan, the dosimetry data were collected from the dose volume histogram (DVH). To account for the dosimetry differences in the hypo-fractionation regimen used in this study and the dosimetry values reported in the literature, usually derived from conventional fractionated regimens (50–60 Gy in 25–30 fractions), the radiation doses were converted to the radiobiological equivalent dose in 2 Gy fractions (EQD2). The α/β value of 2 was considered for late effects on the heart and coronary vessels [8].

All variables were processed as continuous variables, and a descriptive statistical analysis was performed. The distribution was evaluated with the appropriate indexes of centrality and variability, including mean, standard deviation (SD), median, interquartile range (IQR), and range. Spearman’s correlation coefficients were estimated for the variables’ correlation analysis and Bonferroni-adjusted *p*-values were calculated. *p* values < 0.01 were considered statistically significant. The entity of correlation was evaluated assuming that values of Spearman’s coefficient close to ±1 indicated a strong correlation, while values close to 0 indicated no correlation. All statistical analyses were conducted using the open-source R platform (version 4.2.3).

## 3. Results

In all patients’ plans, the dose prescription and the target coverage were successfully achieved, with a mean and SD values for PTV-breast V95% and PTV-boost V95% of 101.4% ± 1.52% and 100.8% ± 0.95%, respectively. The mean and SD values for breast volume was 699.6 cc (±545.5) and for post-operative cavity volume was 110.8 cc ± 75.6 cc. The dosimetry and volume variables are summarized in Table 1.

The mean and SD values for the MHD were 0.69 Gy ± 0.19 Gy (EQD2 0.35 Gy ± 0.1 Gy). The mean and SD values for LAD Dmean and LAD Dmax were 2.20 Gy ± 0.68 Gy (EQD2 1.18 Gy ± 0.35 Gy) and 4.44 Gy ± 1.82 Gy (EQD2 2.55 Gy ± 0.97 Gy), respectively. The mean and SD values for LAD D2%, heart D2% and heart Dmax were, respectively, 3.57 Gy ± 1.25 Gy (EQD2 2 Gy ± 0.65 Gy), 2.35 Gy ± 0.88 Gy (EQD2 1.27 ± 0.45 Gy), and 7.92 Gy ± 6.32 Gy (EQD2 5.01 Gy ± 3.83 Gy).

As shown in Figure 2, Spearman’s correlation analysis revealed a strong positive correlation between the LAD and heart dosimetry variables. In contrast, no strong correlation was observed between the cardiac dose metrics and breast volume, boost volume, or their ratio index.

Regarding the linear regression models between the LAD and heart dosimetry (Table 2 and Figure 3), the analysis showed a linear relationship for each LAD dose variable with heart D2%, described by the following computed equations:LAD Dmean (Gy) = 0.31 + 0.82 × Heart D2% (Gy);LAD D2% (Gy) = 0.63 + 1.23 × Heart D2% (Gy);LAD Dmax (Gy) = 1.02 + 1.38 × Heart D2% (Gy).

**Table 2 cancers-17-01562-t002:** Linear regression models for normalized LAD Dmean, LAD D2%, and LAD Dmax.

**Response Variable = LAD Dmean (Gy)**						
**Explanatory variables**	**Estimate**	**95%CI**	**SE**	**T value**	***p* value for *t*-test**	**R^2^**
Intercept (Gy)	0.31	0.19∓0.44	0.06	4.92	1 × 10^−6^	0.64
Heart Dmean (Gy)	**∓**	**–**	**–**	**–**	**–**	
Heart D2% (Gy)	0.82	0.76–0.87	0.03	30.08	5 × 10^−116^	
Heart Dmax (Gy)	**–**	**–**	**–**	**–**	**–**	
**Response variable = LAD D2% (Gy)**						
**Explanatory variables**	**Estimate**	**95%CI**	**SE**	**T value**	***p* value for *t*-test**	**R^2^**
Intercept (Gy)	0.63	0.43–0.83	0.10	6.13	2 × 10^−9^	0.60
Heart Dmean (Gy)	**–**	**–**	**–**	**–**	**–**	
Heart D2% (Gy)	1.23	1.14–1.31	0.04	28.21	5 × 10^−107^	
Heart Dmax (Gy)	**–**	**–**	**–**	**–**	**–**	
**Response variable = LAD Dmax (Gy)**						
**Explanatory variables**	**Estimate**	**95%CI**	**SE**	**T value**	***p* value for *t*-test**	**R^2^**
Intercept (Gy)	1.02	0.69–1.36	0.17	5.96	5 × 10^−9^	0.41
Heart Dmean (Gy)	**–**	**–**	**–**	**–**	**–**	
Heart D2% (Gy)	1.38	1.24–1.53	0.07	19.09	5 × 10^−62^	
Heart Dmax (Gy)	**–**	**–**	**–**	**–**	**–**	

Abbreviations: LAD = left descending coronary artery, SD = standard deviation, CI = confidence interval, Gy = Gray. Variable selection among heart measures of interest was performed by robust variable selection with exponential squared loss.

**Figure 3 cancers-17-01562-f003:**
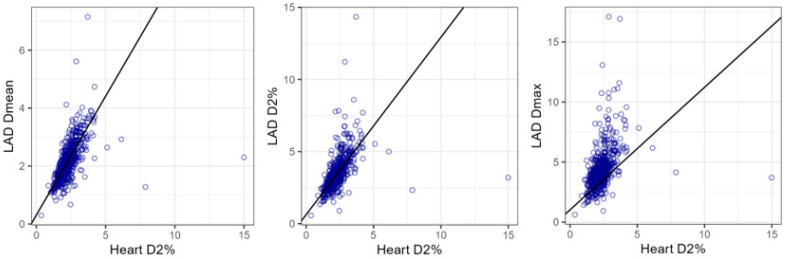
Scatter plots and robust linear regression model lines for LAD Dmean, LAD D2%, and LAD Dmax with heart D2%.

### Comparison with LAD and Heart Dosimetry from Supine Techniques

To minimize the heart and LAD doses delivered during WBRT, several techniques have been developed, including intensity modulation (IMRT), DIBH, and proton therapy.

Most reports in the literature applying a supine setup with free-breathing (FB) photon breast irradiation reported MHD values that generally ranged between 3 and 5 Gy [3,6,8,15,16,17,18,19,20,21,22]. However, Pierce and the Michigan Radiation Oncology Quality Consortium [22] were able to achieve an MHD of 1.32 Gy; the factors for both conventional fractionation and accelerated fractionation that significantly reduced the MHD were the use of DIBH and prone positioning.

The application of DIBH or the prone setup reduced the MHD by 60–70% [15,16,17,18,19,21,22]. Taylor et al. [15] described MHD values of 3.8 Gy for FB supine irradiation, 1.3 Gy for DIBH, and 2.4 Gy for a prone setup. Comparable results were reported in the study by Drost et al. [16] and in the meta-analyses by Lai and colleagues [17,18].

Regarding LAD dosimetry, fewer reports are available. The existing publications document LAD Dmean in the range of 15–20 Gy and LAD Dmax of 35–37 Gy for FB supine breast irradiation [6,8,17,18,19,20,21]. Smyth et al. [19] reported significantly lower LAD Dmean for DIBH compared to FB supine breast RT: LAD Dmean was 10.36 Gy for DIBH versus 20.5 Gy for FB. Notably, the majority of these values were derived from treatment plans with prescription doses of 50–60 Gy in 25–28 fractions to the breast, followed by a sequential boost to the post-operative tumor cavity.

Jimenez et al. [23] reported an MHD of 0.5 Gy RBE (relative biological effectiveness) and LAD Dmean of 1.16 Gy RBE in their phase II study of proton beam RT for 63 patients with left breast cancer, who also required regional nodal irradiation.

The above data from the results of our analysis are summarized in Table 3 and graphically reported in Figure 4.

## 4. Discussion

Studies comparing patients with left-side and right-side breast irradiation have historically shown that left-side patients have a higher incidence of cardiac events and coronary heart disease compared to right-sided ones [1,7,24,25].

While a general consensus exists on MHD constraints [3], no established constraints exist for the LAD [22,26], despite the fact that multiple studies have demonstrated an association of LAD dose with the risk of coronary damage major cardiac events in irradiated left breast cancer patients [5,6,7,8,22,26]. Given the high incidence and prevalence of breast cancer, reducing radiation doses to the heart and LAD carries critical implications for patients’ long-term outcomes [27]. Our group and other groups of investigators modeled how reducing the radiation exposure is particularly beneficial for patients with pre-existing cardiovascular risk factors, where the threshold for harm is lower [26,28,29,30].

This study is unique in that we report comprehensive cardiac dosimetry data for 524 patients with left-side breast cancer who received modern hypo-fractionated WBRT with a concomitant boost to the post-operative cavity, through a consistent, optimized prone irradiation protocol. It reflects a method of prone breast RT that has evolved over two decades, geared at achieving inter- and intra-fraction reproducibility while reaching maximum heart and lung sparing [9,10,11,12,13,14]. The mean values ± SD obtained for MHD, LAD Dmean, and LAD Dmax were 0.69 Gy ± 0.19 Gy (EQD2 0.35 Gy ± 0.1 Gy), 2.2 Gy ± 0.68 Gy (EQD2 1.18 Gy ± 0.35 Gy), and 4.44 Gy ± 1.82 Gy (EQD2 2.55 Gy ± 0.97 Gy), respectively. The reported results demonstrate a significantly lower cardiac exposure compared to other studies and techniques designed at optimizing heart sparing (Table 3 and Figure 4). Moreover, the practical guideline previously reported [12] was validated in the current analysis: at the treatment planning stage, placing the medial edge of the tangent fields at least 2.5 mm from the closest point of the contoured LAD, ensures that LAD Dmax is kept <10 Gy and LAD Dmean < 3.3 Gy [12].

The relevance of an accurate and reproducible prone setup was also confirmed by this study (Figure 1), which, at least in part, explains why other groups have failed to reproduce our results. For instance, Griem et al. [31] compared prone and supine setups in a cohort of 15 patients. Their results showed that the volumes of lung receiving ≥20 Gy and ≥10 Gy were significantly lower in the prone position, without a consistent difference in the irradiated heart volume. However, the paper did not provide figures on fields and dose distributions and failed to describe a standardized protocol for prone positioning, limiting the interpretation of the findings. Chino JP et al. [32] investigated heart displacement due to the effect of gravity by comparing prone diagnostic MR images with supine CT simulation images in 16 cases. As we already reported in a letter following their publication, their positioning of the contralateral arm when prone was likely responsible for inadequate positioning [33]. Vakaet V et al. [34] reported the dosimetry results of a prospective multicentric study of 268 prone and 493 supine patients. The findings confirmed a significantly lower median MHD for left-sided patients, as well as a lower median ipsilateral mean lung dose for all patients treated prone compared to those treated supine. The Ghent University team [35,36,37] has extensively compared supine and prone techniques. Unfortunately, in their prone setup, they used a wedge under the contralateral breast, which results in the axial rotation of the patient and sinking of the chest under the prone table, with gravitational displacement of the heart and greater inclusion within the radiation field. Similarly, Kim DW et al. [38] conducted a comparison of six whole breast radiotherapy (WBRT) techniques, including 3DCRT, VMAT, intensity-modulated proton therapy (IMPT), proton arc therapy (PAT), intensity-modulated carbon-ion therapy (IMCT), and carbon arc therapy, in both supine and prone positions. Their results indicated lower lung exposure in the prone position, while particle beam therapies significantly reduced heart dose regardless of patient positioning. However, once again, the prone setup used in their study did not follow the same positioning constraints applied in our study—specifically, the alignment of the sternum with the inner edge of the immobilization device to maintain a horizontal chest position and prevent sinking or axial rotation.

Other groups have also failed to demonstrate the advantages of prone RT [16,21,22]. The discrepancy with our results is likely to depend on the different techniques used for prone irradiation, compared to our approach that is based on the few simple rules already described [12,13].

Treatment techniques, such as IMRT or volumetric modulated arc therapy (VMAT), have been used to paint the radiation dose in a concave shape around the heart, reducing LAD exposure [22]. These techniques, however, tend to increase the dose exposure to the lung and contralateral breast and the planning requires more time and expertise [39]. In addition, in the United States, many healthcare insurance providers deny coverage for breast IMRT/VMAT. As shown in Appendix A, the prone setup drastically minimizes the breathing excursion of the chest wall without the need for interventions for breath control [40], as compared to DIBH. Limitations of DIBH include patient compliance, particularly among frail and/or elderly patients, longer machine time, and the need for a considerable investment for acquiring the necessary equipment [41].

There is a growing interest in proton therapy as an alternative treatment option to spare the heart, especially in breast cancer patients that require regional nodal irradiation [42]. The largest cohort study of 69 patients was reported by investigators from Massachusetts General Hospital, documenting an MHD of 0.5 Gy (RBE) and an LAD Dmean of 1.16 Gy (RBE) [23]. These values are close to those achieved in our series of patients treated prone with standard photon radiotherapy (Table 3). The RAD-COMP trial [43] compared protons with photons in breast cancer patients to evaluate the benefit in terms of the reduction in major cardiovascular events. Similarly, the Danish Breast Proton Therapy Group (NCT04291378) is recruiting patients to compare photon and proton RT for early breast cancer, with the primary endpoint of reducing cardiac events at 10 years [44]. While the proposed prone setup can be safely and effectively applied to include axillary and supraclavicular nodal irradiation [45], it cannot be applied to target the internal mammary nodes. In our opinion, referral to proton RT in the supine position should be reserved for the subset of patients requiring treatment of the internal mammary nodes.

Conventional photon RT remains the standard for breast RT. WBRT with tangential photon fields in the supine position is still the most commonly adopted technique for adjuvant breast RT, due to its broad availability. Our study demonstrated that a rigorous, heart-sparing protocol of 3D-based prone breast irradiation ensures a cardiac dose reduction of 80–90%, compared to supine FB RT, with comparable dosimetry advantages to those offered by more complex and expensive WBRT techniques (IMRT, VMAT, DIBH, PT) in supine position, reducing costs to the healthcare system through an accessible and sustainable therapeutic option. In an original prospective study published in *JAMA* in 2012 [46], we reported the results of prone versus supine positioning in 400 patients with early-stage breast cancer who consented to undergo simulation in both positions. The study showed that in most cases prone positioning significantly reduced radiation exposure to the heart and lungs. Specifically, lung dose was reduced by 91.1% in left-sided and 86.2% in right-sided cancers, while heart dose was reduced by 85.7% in left-sided cases, independently of breast size.

A limitation of the current study is its retrospective nature and the fact that it represents the experience of a single institution. Nevertheless, prone WBRT is a feasible and effective approach for routine clinical practices [9,10,11,12,13,14,29,40,46], demonstrating significant reductions in radiation exposure to the heart and lungs, along with improved setup reproducibility and patient comfort assured by the modern breast boards. Our team has also successfully trained many radiation oncologists to correctly adopt the described approach of prone breast RT, rendering it available at selected centers worldwide.

## 5. Conclusions

This study demonstrates that, in patients with left-side breast cancer, the adoption of a careful heart-sparing protocol of prone hypo-fractionated WBRT with a concomitant boost to the post-operative cavity results in optimal dose-sparing of the heart and the LAD, regardless of individual body conformation and treatment volumes, without compromising target coverage. This approach can be easily adopted at any RT-based facility with the potential for offering a safe and sustainable care path for breast cancer treatment globally. As a future direction, a collaborative project with our NYU colleagues is currently underway to investigate late cardiovascular effects in women with breast cancer who were treated with prone WBRT at that institution between 2000 and 2010, by the same research group.

## Figures and Tables

**Figure 1 cancers-17-01562-f001:**
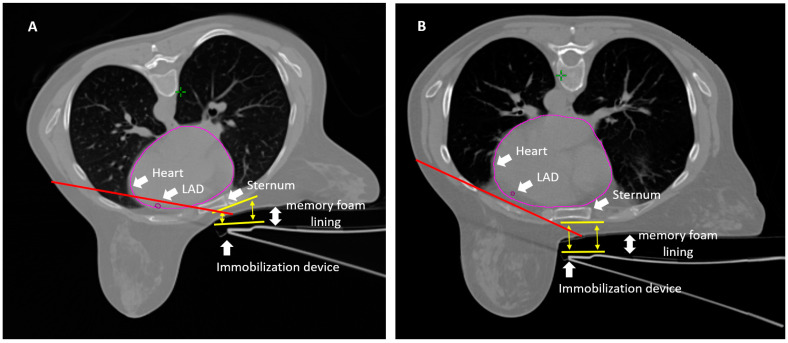
Example of an incorrect (**A**) and a correct (**B**) prone setup.

**Figure 2 cancers-17-01562-f002:**
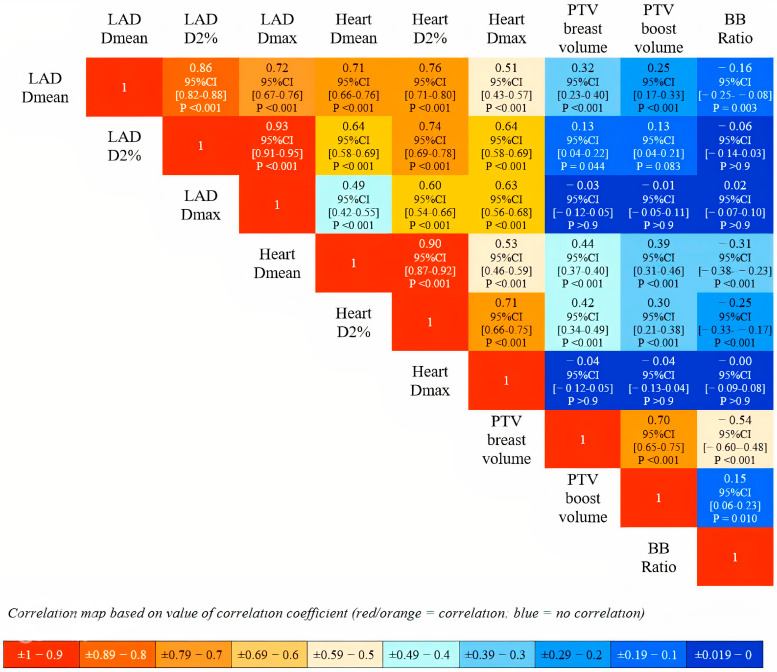
Spearman’s correlation analysis with Bonferroni-adjusted *p*-values.

**Figure 4 cancers-17-01562-f004:**
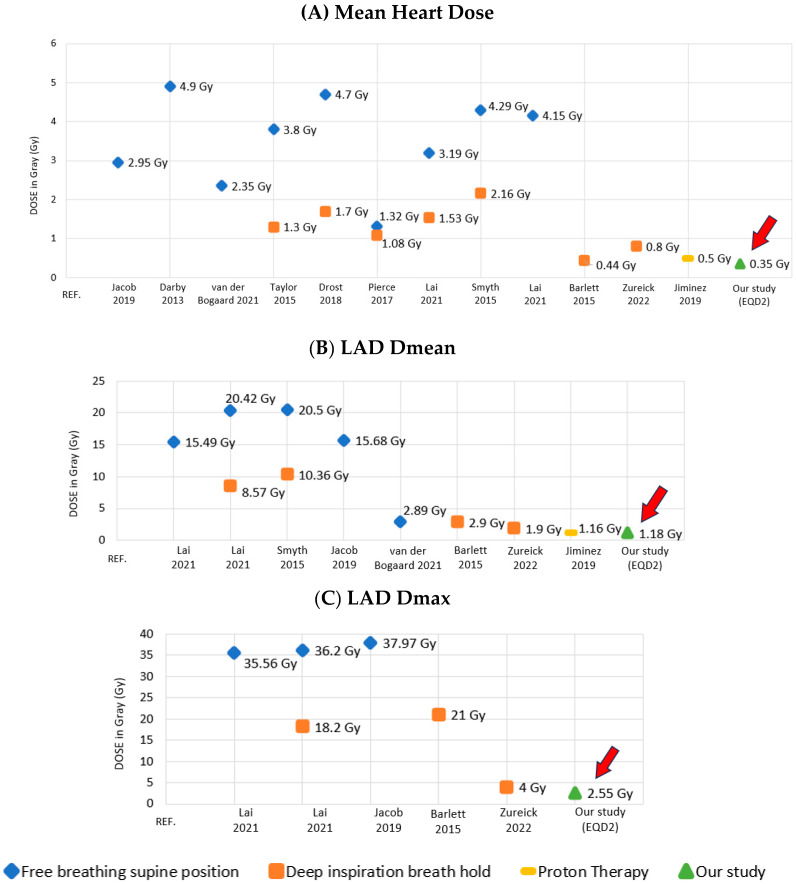
Dosimetry of mean heart dose (**A**), LAD Dmean (**B**), and LAD Dmax (**C**) among different setups and techniques for left breast irradiation [3,6,8,12,15,16,18,19,20,22,23].

**Table 1 cancers-17-01562-t001:** Dosimetry and volumes variables.

Structure	Variable	Mean	SD	Median	IQR	Range
**Heart**						
	MHD (Gy)	0.69	0.19	0.67	0.56–0.78	0.12–1.88
	*EQD_2 (Gy)_*	*0.35*	*0.10*	*0.34*	*0.29–0.40*	*0.06–1.00*
	Dmax (Gy)	7.92	6.32	5.02	3.71–8.75	0.57–39.82
	*EQD_2 (Gy)_*	*5.01*	*3.83*	*2.93*	*2.08–5.65*	*0.29–46.34*
	D2% (Gy)	2.35	0.88	2.27	1.86–2.65	0.35–7.87
	*EQD_2 (Gy)_*	*1.27*	*0.45*	*1.22*	*0.99–1.44*	*0.18–4.97*
**LAD**						
	Dmean (Gy)	2.20	0.68	2.13	1.74–2.56	0.29–5.61
	*EQD_2 (Gy)_*	*1.18*	*0.35*	*1.14*	*0.92–1.39*	*0.15–3.33*
	Dmax (Gy)	4.44	1.82	4.09	3.37–5.07	0.61–16.92
	*EQD_2 (Gy)_*	*2.55*	*0.97*	*2.32*	*1.87–2.96*	*0.31–13.23*
	D2% (Gy)	3.57	1.25	3.32	2.83–4.08	0.57–11.22
	*EQD_2 (Gy)_*	*2.00*	*0.65*	*1.84*	*1.55–2.32*	*0.29–7.71*
**PTV breast**						
	V95% (%)	101.4	1.52	101.3	100.5–101.9	99.61–107.58
**Breast volume**						
	Volume (cc)	699.6	545.5	560.2	320.8–901.1	41.54–3747.50
**PTV boost**						
	V98% (%)	100.8	0.95	100.8	100.5–101.1	99.51–104.73
**Post-operative cavity volume**						
	Volume (cc)	110.8	75.6	94.88	61.4–141.8	9.53–701.04

**Table 3 cancers-17-01562-t003:** Summary of data for mean heart dose, LAD Dmean, and LAD Dmax for left breast cancer adjuvant radiotherapy reported in the literature and from our cohort, categorized by different techniques.

Author/ Publication Year	Evaluation Period	Study Design	Treatment Planning, Field Type, Dose/Fractions	Sample Size (Patients/Studies)	MHD for FB Supine Position	MHD for DIBH Supine Position	MHD for Prone Position	LAD Dmean for FB Supine Position	LAD Dmean for DIBH Supine Position	LAD Dmean for Prone Position	LAD Dmax for FB Supine Position	LAD Dmax for DIBH Supine Position	LAD Dmax for Prone Position
Darby 2013	1958–2001 (Sweden) 1977–2000 (Denmark)	Large retrospective study	Oblique anterior, direct anterior, tangential pair; 40–52 Gy/20–28 fx	2168 patients	4.9 Gy ^ (SD 4.4)	-	-	-	-	-	-	-	-
Taylor 2015	2003–2013	Systematic Review	3D conformal tangents; IMRT, 40–50 Gy/15–28 fx	NA (149 studies)	3.8 Gy * (SE 0.2)	1.3 Gy * (SE 0.1)	2.4 Gy * (SE 0.5)	-	-	-	-	-	-
Smyth 2015	1966–2014	Systematic Review	3D conformal tangents; IMRT; 50 Gy/25 fx, 40–42 Gy/15–16 fx	268 patients (10 studies)	4.29 Gy *	2.16 Gy *	-	20.5 Gy *	10.36 Gy *	-	-	-	-
Bartlett 2015	2013–2014	Randomized study	3D conformal tangents; 40 Gy/15 fx	34 patients	-	0.44 Gy ^ (95% CI 0.38–0.51)	0.66 Gy ^ (95% CI 0.61–0.71)	-	2.9 Gy ^ (95% CI 1.8–3.9)	7.8 Gy ^ (95% CI 6.4–9.2)	-	21.0 Gy ^ (95% CI 15.8–26.2)	36.8 Gy ^ (95% CI 35.2–38.4)
Pierce 2017	2012–2015	Prospective observational large study	3D conformal tangents; conventional + accelerated fractionation	4688 patients	1.32 Gy ^ (Range 1.20–1.45)	1.08 Gy ^ (Range 1.03–1.14)	0.89 Gy ^ (Range 0.79–1.00)	-	-	-	-	-	-
Drost 2018	2014–2017	Systematic Review	2D, 3D conformal tangents, IMRT; 50 Gy/25–28 fx, 40–42 Gy/15–16 fx	11,545 patients (99 studies)	4.7 Gy * (Range 0.1–18.7)	1.7 Gy * (Range 0.4–4.8)	2.3 Gy * (Range 0.7–4.2)	-	-	-	-	-	-
Jacob 2019	2015–2017	Prospective study	3D conformal tangents; 47–50 Gy/20–25 fx	89 patients	2.95 Gy ^ (SD 1.49)	-	-	15.68 Gy ^ (SD 8.13)	-	-	37.97 Gy ^ (SD 15.93)	-	-
Lai 2020	2011–2018	Meta-analysis	NA; 50 Gy/25 fx	1019 patients (12 studies)	3.19 Gy * (SD 1.8)	1.53 Gy * (SD 0.87)	-	20.42 Gy * (SD 10.15)	8.57 Gy * (SD 6.68)	-	36.2 Gy * (SD 11.52)	18.2 Gy * (SD 13.99)	-
Lai 2021	2007–2020	Meta-analysis	NA; 50 Gy/25 fx, 40–42 Gy/15–16 fx	751 patients (19 studies)	4.15 Gy * (SD 1.93)	-	3.18 Gy * (SD 1.45)	15.49 Gy * (SD 6.8)	-	12.29 Gy * (SD 6.8)	35.56 Gy * (SD 11.1)	-	26.76 Gy * (SD 12.48)
van den Bogaard 2021	2005–2008	Large retrospective study	3D conformal tangents; 50 Gy/28 fx	910 patients	2.35 Gy ^ (Range 0.51–15.25)	-	-	2.82 Gy ^ (Range 0.41–59.33)	-	-	-	-	-
Zureick 2022	2012–2018	Large retrospective study	3D-conformal tangent fields; 50 Gy/25–28 fx; 42 Gy/16 fx	375 patients	-	0.8 Gy ^ (IQR 0.6–1.1)	-	-	1.9 Gy ^ (IQR 1.4–3.2)	-	-	4.0 Gy ^ (IQR 3–11)	-
# Jimenez 2019	2011–2019	Phase II study	3D-conformal proton therapy; 50.4 Gy (RBE) in 28 fx (chest wall) 45.0 Gy (RBE) in 25 fx (breast)	63 patients	0.5 Gy (RBE) (Range 0.1–1.7)	-	-	1.16 Gy (RBE) (Range 0.09–12)	-	-	-	-	-
Our study 2024	2016–2023	Large Retrospective study	3D conformal tangents; IMRT; 40.5 Gy/15 fx + SIB	524 patients	-	-	0.69 Gy ^ (SD 0.19) EQD2 0.35 Gy ^ (SD 0.10)	-	-	2.20 Gy ^ (SD 0.68) EQD2 1.18 Gy ^ (SD 0.35)	-	-	4.44 Gy ^ (SD 1.82) EQD2 2.55 Gy ^ (SD 0.97)

* Dose defined as the arithmetic mean of doses reported by studies. ^ Dose defined as mean or median value. # Proton therapy with regional node irradiation. Abbreviations: NA = not available; MHD = mean heart dose; FB = free breathing; DIBH = deep inspiration breath hold; LAD = left descending coronary artery; Dmean = mean dose; Dmax = maximum dose; SE = standard error; SD = standard deviation; CI = confidence interval; Gy = Gray, EQD2 = 2-Gy Equivalent Dose; fx = fractions; and SIB = simultaneous integrated boost.

## Data Availability

The data presented in this study are available on request from the corresponding author due to privacy and legal reasons.

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
