# Peer review of "Heart and Left Anterior Descending Coronary Artery (LAD) Exposure from Hypo-Fractionated Whole Breast Radiotherapy with a Prone Setup"

_cancers, 2025, doi:10.3390/cancers17091562_

Round 1

Reviewer 1 Report

Comments and Suggestions for Authors

This article is dealing with cardiac doses from breast cancer radiotherapy with a prone set-up.  The article is well written and gives heart and LAD doses from a large group of patients  (n=524). This is the strong point of the work.

However, the article has some shortcomings. The research part of this study is rather limited and the methodology is simplistic. The irradiation technique has been previously described. The authors simply collected the values of certain dosimetric parameters using DVHs extracted from the patients’ plans. No attempts were made to analyze these cardiac doses (dose variation, dose vs breast volume, …).  The dose values were then compared with those reported in the literature. Their conclusion, related to the superiority of prone setup for heart sparing compared to supine position, has been discussed.

Author Response

REVIEWER 1

This article is dealing with cardiac doses from breast cancer radiotherapy with a prone set-up.  The article is well written and gives heart and LAD doses from a large group of patients (n=524). This is the strong point of the work.

However, the article has some shortcomings. The research part of this study is rather limited, and the methodology is simplistic. The irradiation technique has been previously described. The authors simply collected the values of certain dosimetry parameters using DVHs extracted from the patients’ plans. No attempts were made to analyze these cardiac doses (dose variation, dose vs breast volume, …).  The dose values were then compared with those reported in the literature. Their conclusion, related to the superiority of prone setup for heart sparing compared to supine position, has been discussed.

Thank you for your review and your comments.

In response to your point, we have re-included the more technical part of the paper, that we had removed to ease the lecture for a larger audience than radiation oncologists and physicists. It is now added, from line 186 to 205, and it includes two additional figures and a table.

Reviewer 2 Report

Comments and Suggestions for Authors

This study evaluates the impact of prone positioning on radiation dose reduction to the heart and left anterior descending coronary artery (LAD) in patients with left-sided breast cancer undergoing whole breast radiotherapy (WBRT). The findings demonstrate that prone positioning significantly lowers cardiac radiation exposure compared to supine techniques, potentially reducing long-term cardiac toxicity. This work highlights the dosimetric advantages of prone WBRT, supporting its role as a safer and cost-effective approach in breast cancer treatment. However, I have some suggestions before accepting it, please arrange a minor revision:

  1. More detailed discussion on the clinical significance of the observed dosimetry reduction in terms of long-term cardiac toxicity should be discussed.
  2. The comparison with supine techniques is primarily based on literature data. It would be helpful to discuss potential variability in patient selection, contouring methods, or treatment planning across studies.
  3. The methodology section should clarify whether patient anatomical variations (e.g., breast size, BMI) influenced dose distribution.
  4. Including a brief discussion on the feasibility of implementing prone WBRT in routine clinical practice (e.g., patient comfort, setup time, reproducibility) would enhance the practical implications of the study.
  5. If possible, providing outcome data beyond dosimetry, such as preliminary toxicity or treatment efficacy results, would strengthen the clinical relevance.
  6. Consider discussing the implications of these findings in the context of modern techniques such as deep inspiration breath-hold (DIBH), which is commonly used for heart sparing in the supine position.

Author Response

REVIEWER 2

This study evaluates the impact of prone positioning on radiation dose reduction to the heart and left anterior descending coronary artery (LAD) in patients with left-sided breast cancer undergoing whole breast radiotherapy (WBRT). The findings demonstrate that prone positioning significantly lowers cardiac radiation exposure compared to supine techniques, potentially reducing long-term cardiac toxicity. This work highlights the dosimetry advantages of prone WBRT, supporting its role as a safer and cost-effective approach in breast cancer treatment. However, I have some suggestions before accepting it, please arrange a minor revision:

  1. More detailed discussion on the clinical significance of the observed dosimetry reduction in terms of long-term cardiac toxicity should be discussed.
  2. The comparison with supine techniques is primarily based on literature data. It would be helpful to discuss potential variability in patient selection, contouring methods, or treatment planning across studies.
  3. The methodology section should clarify whether patient anatomical variations (e.g., breast size, BMI) influenced dose distribution.
  4. Including a brief discussion on the feasibility of implementing prone WBRT in routine clinical practice (e.g., patient comfort, setup time, reproducibility) would enhance the practical implications of the study.
  5. If possible, providing outcome data beyond dosimetry, such as preliminary toxicity or treatment efficacy results, would strengthen the clinical relevance.
  6. Consider discussing the implications of these findings in the context of modern techniques such as deep inspiration breath-hold (DIBH), which is commonly used for heart sparing in the supine position.

Thank you for your thorough review of our work and for the valuable suggestions. Below we describe how we believe we have addressed each of your points.

Specifically:

  1. Discussion is now extended (page 11) to stress the relevance of the findings, from line 250 to 259.

  1. We are now more clearly referring to our JAMA paper that in fact reported such a comparison in a prospective cohort of 400 women (200 with right-site breast cancer and 200 with left-site breast cancer) simulated in both positions and then treated in the position that best spared normal organs while assuring coverage of the index breast (page 12, from line 342 to 348).

  1. In results we are now specifying the relationship between breast size and dosimetry (pages 5-7, from line 186 to 205).

  1. In discussion we are now expanding the paragraph about the universal applicability of the prone set up (page 13, from line 350 to 354).

  1. While it is currently impossible to correlate normal tissue sparing with late effects in the cohort reported, an independent project, in collaboration with NYU is ongoing to report late effects among women treated prone at that institution from 2000-2010, by the same investigators. Page 13 (from line 364 to 367) refers to such an ongoing effort.

  1. discussion is now inclusive of a comparison of prone RT with DIBH (equipment cost, time on the table, compliance etc.) (page 12, from line 302 to 306).

Again, thank you for your valuable advice.

Reviewer 3 Report

Comments and Suggestions for Authors

This research shows advantages in Heart and LAD doses sparing in free breathing prone position versus supine using 3D technique. However  there are growing evidences  not confirming this gain. Griem KL  in Red Journal  2003 demonstrated no variation of irradiated heart volume in prone position vs supine; Chino JP in Red Journal 2008 showed  systematic displacement of the lateral and superior aspect of the heart closer to the chest wall in the prone vs. supine position (mean displacement 19 mm (95% confidence interval 13.7–25.1 mm, p < 0.001) with   negative consequences in situations in which the high-risk target tissues include the chest wall or deep breast . Similarly Vokaet V  confirmed no advantages in heart and LAD sparing in prone crawl position.  A dosimetric analysis by Kim DW 2024 has shown pros and contra in prone and supine among six irradiation techniques. In the metaanalysis of Lai J ( ref.18) the novelty was found in Prone DIBH studies.  Prone DBIH and repeated prone DBIH which have  shown a significant gain in heart and LAD sparing as in the paper of Deseyne P (2022) and Speleers(2021) with photons and protons.  

Author Response

REVIEWER 3

This research shows advantages in Heart and LAD doses sparing in free breathing prone position versus supine using 3D technique. However, there are growing evidence not confirming this gain. Griem KL  in Red Journal  2003 demonstrated no variation of irradiated heart volume in prone position vs supine; Chino JP in Red Journal 2008 showed  systematic displacement of the lateral and superior aspect of the heart closer to the chest wall in the prone vs. supine position (mean displacement 19 mm (95% confidence interval 13.7–25.1 mm, p < 0.001) with   negative consequences in situations in which the high-risk target tissues include the chest wall or deep breast . Similarly, Vakaet V confirmed no advantages in heart and LAD sparing in prone crawl position.  A dosimetry analysis by Kim DW 2024 has shown pros and contra in prone and supine among six irradiation techniques. In the metanalysis of Lai J (ref.18) the novelty was found in Prone DIBH studies.  Prone DBIH and repeated prone DBIH which have shown a significant gain in heart and LAD sparing as in the paper of Deseyne P (2022) and Speleers (2021) with photons and protons.  

Thank you for your comments and your concerns. In recognition of your important comments, we have now modified the manuscript to discuss the references you suggested. In our opinion, the discrepancies with our findings described in most of manuscripts quoted result from an inaccurate/inadequate application of the prone set-up, as now discussed in multiple sites of the manuscript. We sincerely hope that the adoption of the simple guidelines we have listed will enable a wider successful application of the prone technique we propose.

Round 2

Reviewer 3 Report

Comments and Suggestions for Authors

The paper shows the advantages in dosimetry in prone position and hypofractionated radiotherapy ; however it is a solution among others trying to spare heart and LAD like DIBH which is the standard for left side breast cancer in supine position. Thus this is an option for women not suitable for supine DIBH and viceversa. Moreover the set-up is crucial in this prone solution